# Effect of Fragment 1 on the Binding of Epigallocatechin Gallate to the PD-L1 Dimer Explored by Molecular Dynamics

**DOI:** 10.3390/molecules28237881

**Published:** 2023-11-30

**Authors:** Yan Guo, Yilin Guo, Zichao Guo, Boping Liu, Jianguo Xu

**Affiliations:** 1College of Food Science, Shanxi Normal University, Taiyuan 030031, China; guoyan@sxnu.edu.cn (Y.G.); 222420010@sxnu.edu.cn (Y.G.); 2230030139@sxnu.edu.cn (Z.G.); 2Key Laboratory for Bio-Based Materials and Energy of Ministry of Education, College of Materials and Energy, South China Agricultural University, Guangzhou 510630, China

**Keywords:** PD-1/PD-L1 pathway, epigallocatechin gallate, fragment 1, cancer immunotherapy, molecular dynamics simulation

## Abstract

Blocking the interaction between programmed cell death-1 (PD-1) and programmed cell death-ligand 1 (PD-L1) by directly targeting the PD-L1 dimer has emerged as a hot topic in the field of cancer immunotherapy. Epigallocatechin gallate (EGCG), a natural product, has been demonstrated binding to the PD-L1 dimer in our previous study, but has a weaker binding capacity, moreover, EGCG is located at the end of the binding pocket of the PD-L1 dimer. The inhibitor fragment 1 (FRA) lies at the other end. So, we proposed that the introduction of FRA might be able to improve the binding ability. To illuminate this issue, molecular dynamics (MD) simulation was performed in the present study. Binding free energy calculations show that the binding affinity is significantly increased by 17 kcal/mol upon the introduction of FRA. It may be due to the energy contributions of emerging key residues _A_Tyr56, _A_Met115, _B_Tyr123, _A_Ile54 and the enhanced contributions of initial key residues _A_Tyr123 and _B_Val68. Binding mode and non-bonded interaction results indicate that FRA_EGCG (EGCG in combination with FRA) binds to the C-, F- and G-sheet of the PD-L1 dimer. Importantly, the introduction of FRA mainly strengthened the nonpolar interactions. The free energy landscape and secondary structure results further show that FRA_EGCG can interact with the PD-L1 dimer more stably. These data demonstrated here provide the theoretical basis for screening two or more natural products with additive inhibitory effect on this pathway and therefore exerting more effective anticancer immunity.

## 1. Introduction

PD-1 is a co-inhibitory receptor broadly expressed on the surface of immune cells, such as activated T cells, B cells and natural killer cells [1,2]. Interaction of PD-1 with its ligand, PD-L1, when expressed on cancer cells triggers exhaustion of immune cells [2,3,4,5,6]. Therefore, blockade of PD-1/PD-L1 interaction is a promising approach to restoring T cells for removing cancer cells [7]. Indeed, several monoclonal antibodies (mAbs) have been approved to treat multiple types of cancer, and some of them are undergoing clinical trials [7,8,9,10]. However, mAbs therapies display several disadvantages, such as poor tumor permeability and no oral bioavailability [7,11,12]. Thus, focusing on the development of small-molecule inhibitors is a more appealing approach [13]. Moreover, superior membrane permeability, good stability and other features make small molecules more suitable for clinical treatments in the future [14]. 

Numerous natural products such as polyphenols have attracted considerable attention due to their diverse biologic activities, low toxicity and side effects, and potential immunomodulatory effect in cancer cells [15,16,17,18,19]. Indeed, several polyphenols have been reported as blockades of the PD-1/PD-L1 axis [18,20,21,22,23,24]. Notably, the findings in our previous work provided computational insights that EGCG could directly target PD-L1 dimerization, though lower relative binding capacities (Δ*G* = −20.31 kcal/mol) [25]. Analysis of the binding mode revealed that EGCG does not completely fill the entire cavity of the PD-L1 dimer. In addition, Perry et al. identified 36 small-molecule inhibitors of this pathway based on a fragment-based method [26]. More importantly, EGCG and FRA are located at two different ends of the cavity. Hence, we proposed a scientific hypothesis that the introduction of FRA is a good approach for improving the binding capacity of EGCG. Thus, it is of great importance to investigate the underlying interaction mechanism between the PD-L1 dimer and FRA_EGCG in detail.

With the help of fast-growing computational power, molecular simulations have offered a large amount of useful information on the atomic level of protein–ligand systems [27,28]. Hence, taking advantage of the availability of the crystal structures of EGCG, FRA and the PD-L1 dimer in complex with BMS-200, molecular docking was first performed to acquire the complex systems (Figure 1 and Figure 2) [29,30]. Then, MD simulation was carried out to study the binding properties of EGCG, FRA and FRA_EGCG with the PD-L1 dimer [29]. Afterwards, binding free energy calculation was undertaken by utilizing the molecular mechanics Poisson–Boltzmann surface area (MM-PBSA) approach to assess the relative binding affinities of the complex systems [29,31]. Finally, analyses of principal component and secondary structure were used to explore the internal dynamic characteristics of the PD-L1 dimer [32]. Briefly, this study aims to provide a significant theoretical hint to discover two or more potent natural products jointly for cancer immunotherapy.

## 2. Results and Discussion

### 2.1. Root-Mean-Squared Deviation

The PD-L1 dimer/EGCG, PD-L1 dimer/FRA and PD-L1 dimer/FRA_EGCG complex systems were produced by the molecular docking method which has been validated in our previous studies [25,33]. Independent MD simulations were then implemented to assess their binding properties. The root-mean-squared deviation (RMSD) values over the residues within 20 Å of small molecules were first measured to ensure the systems reach an equilibrium state during 150 ns of simulation time [33]. As illustrated in Figure 3a, the overall MD trajectories of the complex systems showed similar trends, among which the FRA_EGCG system can be quickly equilibrated within 2 ns and focuses on 2.30 Å, while the FRA and EGCG systems reach convergence at about 30 ns and 5 ns with fluctuations of 2.90 and 3.11 Å, respectively. In comparison, the structures in the trajectories excluding small molecules significantly differed from those in the complex systems, and fluctuated significantly around 3.00–5.35 Å between 30 and 60 ns. These results illustrated that small molecules presumably affect the conformation and stability of the PD-L1 dimer, thereby leading to a decrease in RMSD values. Generally, all MD trajectories could be applied for subsequent analyses.

### 2.2. Root-Mean-Square Fluctuation

The root-mean-square fluctuation (RMSF) was employed to monitor the structural fluctuation in each residue of the systems, enabling to identify more stable and flexible regions. As illustrated in Figure 3b,c, the overall fluctuation pattern of the dimer system appears similar as observed for small molecules bound to the PD-L1 dimer in timescale simulations. It was evident that residues in the N-terminal, C-terminal and loop showed higher RMSF values, which reached up to ~5 Å in the dimer system. Moreover, β-sheets including sheets A–G represented more stable behaviors and the RMSF values only reached ~2 Å. Of course, differences are also obvious. Fluctuations of the complex systems are alleviated compared to those observed in the dimer system. Notably, the residues 54–59, 110–117 and 121–124 of the complex systems (existing in the C-, F- and G-sheet, respectively) were proved to fluctuate less and only reached a maximum RMSF of 1 Å. These observations indicated that the PD-L1 dimer undergoes more stable rearrangement due to the binding of small molecules, which is in agreement with the findings of the RMSD results.

### 2.3. Binding Free Energy

To qualitatively characterize the binding affinities of the PD-L1 dimer to small molecules, the binding free energies (Δ*G*) were calculated using the MM-PBSA approach based on the 300 frames extracted from the last 30 ns stable trajectories of each MD simulation and summarized in Table 1.

The Δ*G* of the PD-L1 dimer in complex with FRA, EGCG and FRA_EGCG are −24.27 ± 0.27, −20.31 ± 0.35 and −37.51 ± 0.43 kcal/mol in sequence, suggesting that FRA_EGCG possessed great stronger interaction energy with the PD-L1 dimer compared with EGCG. Nevertheless, the Δ*G* of the dimer system was positive (36.11 ± 0.89 kcal/mol), indicating that PD-L1 can hardly be spontaneously dimerized at all, and these small molecules are crucial to the dimerization of PD-L1. Along with the total binding free energy, the contributions of each individual term (Δ*E*_vdw_, Δ*E*_ele_, Δ*E*_PB_, and Δ*E*_SA_) were also derived in the present study. As displayed in Table 1, the contribution of Δ*E*_vdw_ (−74.75 ± 2.19 kcal/mol for the FRA_EGCG system, −37.14 ± 0.51 kcal/mol for the FRA system and −42.88 ± 1.70 kcal/mol for the EGCG system) were much higher than that of other interaction energies. Moreover, the Δ*E*_ele_ and Δ*E*_SA_ for the complex systems were −32.17 ± 2.29, −3.43 ± 0.13, −23.14 ± 3.14 kcal/mol and −7.48 ± 0.28, −3.14 ± 0.02, −4.73 ± 0.16 kcal/mol, respectively. Conversely, the Δ*E*_PB_ of the complex systems were calculated to be 76.89 ± 2.94, 19.43 ± 0.83 and 50.43 ± 3.76 kcal/mol, indicating that the polar solvation energy term was unfavorable to the binding of small molecules. This term was increased with the size of the small molecules [34]. In general, the total nonpolar binding free energy was found to be the major driving force for their binding affinities, especially the FRA_EGCG system with the value −82.23 ± 2.35 kcal/mol. The electrostatic interactions can also make additional contributions to the binding for the complex systems. Notably, FRA mainly enhanced the nonpolar energies of the FRA_EGCG system, and the Δ*E*_ele_ of this system mostly originated from the interactions between the PD-L1 dimer and EGCG. This observation may be related to the aromatic rings of FRA and hydroxyl groups of EGCG. These features should be taken into account when screening and discovering natural small-molecule inhibitors of the PD-1/PD-L1 pathway.

### 2.4. Per-Residue Energy Decomposition

The total binding free energy was further decomposed into per-residue contribution of the PD-L1 dimer to identify residues that play a crucial role in the binding of small molecules. As shown in Figure 4, the energy contributions of five residues (_A_Tyr123, _B_Ile54, _B_Tyr56, _B_Val68, and _B_Met115) to binding with EGCG were found to be more than −1.00 kcal/mol with their respective ΔG values of −1.26 ± 0.12, −2.50 ± 0.40, −1.99 ± 0.04, −1.09 ± 0.36, and −1.30 ± 0.55 kcal/mol. The residue contributions for the FRA system were mainly attributed to _A_Tyr56, _A_Met115, and _B_Tyr123, while the major energy contributions of the FRA_EGCG system originated from nine residues (_A_Ile54, _A_Tyr56, _A_Met115, _A_Tyr123, _B_Ile54, _B_Tyr56, _B_Val68, _B_Met115, and _B_Tyr123) with their ΔG values calculated to be −1.29 ± 0.41, −1.82 ± 0.21, −1.49 ± 0.68, −1.53 ± 0.47, −2.27 ± 1.12, −1.87 ± 0.15, −1.82 ± 0.77, −1.12 ± 0.28, and −1.32 ± 0.04 kcal/mol in sequence. It was observed that these key binding residues exhibited fairly low RMSF values, under 1 Å in the complex systems, whereas the RMSF values of other residues, aside from that of the active site, were generally high (Figure 3b,c). Furthermore, compared with the FRA system, residues _A_Ile54, _A_Tyr123, _B_Ile54, _B_Tyr56, _B_Met115, _B_Val68, and _B_Tyr123 devoted more to the FRA_EGCG system. When compared to the EGCG system, the emerging key residues in the FRA_EGCG system consist of _A_Met115, _B_Tyr123, _A_Tyr56, and _A_Ile54, among which the first three are key residues of the FRA system. Similarly, the importance of these key residues for the binding to both synthetic and natural small-molecule inhibitors have been highlighted by our group and other researchers [6,21,25,33,35,36]. Of course, the energy contributions of residues including _A_Ser117, _A_Tyr123, _B_Val68, _B_Val76, and _B_Ala121 were also strengthened. It was indicated that the introduction of FRA not only increases the numbers of key residues at the binding interface, but also enhances the contributions of some other residues to binding affinities. More importantly, the key residues of the FRA_EGCG system are located in the primary hot spot of the druggability of the PD-L1 surface [37]. Thus, we can infer that EGCG and FRA are effective antagonists of the PD-1/PD-L1 pathway, and FRA_EGCG could exert more efficient inhibitory effect on this pathway.

### 2.5. Contact Numbers

To characterize the binding regions of small molecules on the PD-L1 dimer, the average contact numbers between small molecules and individual residues were calculated (Figure 5). Herein, the same standard was utilized to identify important residues as described in our previous studies [25,33].

As shown, EGCG preferentially bound to residues Phe19, Thr20, Ile54, Val55, Tyr56, Gln66, Val68, Met115, Ile116, Ser117, Ala121, Asp122, Tyr123, and Lys124. FRA preferred to interact with Ile54, Tyr56, Met115, Ile116, Ser117, Ala121, Asp122, and Tyr123. Meanwhile, FRA_EGCG exhibited strong preferential interactions with residues Phe19, Thr20, Ile54, Val55, Tyr56, Gln66, Val68, His69, Asp73, Leu75, Lys76, Met115, Ile116, Ser117, Ala121, Asp122, Tyr123, and Lys124. Hence, compared with the EGCG system, emerging important residues include _B_Asp73, _B_Leu75, _B_Lys76, _A_Ile54, _A_Tyr56, _A_Met115, _B_Ala121, and _B_Tyr123 in the FRA_EGCG system. In brief, these small molecules mainly interacted with the C sheet (residues 54–56), F sheet (residues 115–117) and G sheet (residues 121–124) of the PD-L1 dimer. Moreover, both EGCG and FRA_EGCG bound to additional regions including the N-terminal (residues 19–20) and C’ sheet (residues 66–68). FRA_EGCG bound to the C’D loop (residues 69–72) and α-helix (residues 73–76). These observations match well with the calculated binding free energies in Table 1 and Figure 4.

### 2.6. Non-Bonded Interactions

Quantitative investigation of H bonds was then performed to scrutinize the interaction patterns of the complex systems. The occupancies maintained for >50% along the entire MD simulation were considered as stable H bond interactions [38]. As shown in Table 2, EGCG kept stable H bonds with the O atom of _A_Ala121, O atom of _A_Phe19, and N atom of the negatively charged residue _A_Asp122, with their occupancies of 80.40%, 59.47%, and 59.80% in the EGCG system. EGCG maintained H bonds with the O atoms of _A_Asp122 and polar residue _B_Gln66 in the FRA_EGCG system, and the occupancy values (>80%) further proved their stabilities, implying that these residues were tightly bonded in the pocket (Table 3). Nevertheless, H-bond interactions between FRA and the PD-L1 dimer were not observed in both the FRA_EGCG and FRA systems. In short, the H bond interactions of the FRA_EGCG system is similar to the EGCG system though the occupancies were different, which may be related to multiple hydroxyl groups of EGCG.

The detailed binding modes of small molecules with the PD-L1 dimer are displayed in Figure 6. As illustrated, these small molecules were located in a cylindrical tunnel between _A_PD-L1 and _B_PD-L1. The pocket of the EGCG system was surrounded by residues _A_Phe19, _A_Thr20, _A_Ala121, _A_Asp122, _A_Lys124, _B_Gln66, _A_Tyr123, _B_Ile54, _B_Tyr56, _B_Val68, and _B_Met115, in which the last five residues formed hydrophobic interactions with EGCG. The pocket of the FRA system was observed to be surrounded by the sidechains of residues _A_Tyr56, _B_Tyr123, _B_Ile54, _A_Met115, _B_Met115, and _A_Ala121, among which the last four residues formed hydrophobic interactions with FRA. The binding pocket of the FRA_EGCG system was surrounded by residues _A_Tyr56, _A_Met115, _B_Tyr123, _A_Ile54, _A_Ala121, _A_Ser117, _A_Asp122, _A_Lys124, _B_Gln66, _B_His69, _B_Met115, _B_Ile54, _B_Tyr56, _B_Val68, _B_Val76, _A_Ala121, and _A_Tyr123. In detail, FRA formed hydrophobic interactions with the first three residues, and such interactions were also observed between the sidechains of the last six residues and ECCG. Briefly, introduction of FRA mainly strengthened the nonpolar interactions, the significant role of which has been emphasized in our previous studies and other studies [6,25,33,35,36]. On the other hand, it is conceivably indicated that two or more natural inhibitors may exert additive inhibitory effects on the PD-1/PD-L1 pathway. Indeed, natural products in combination with antibodies have been reported to show remarkable synergistic inhibition against the pathway [20]. 

### 2.7. Cross-Correlation Matrix Analysis

To further explore the internal dynamical behavior of the PD-L1 dimer, cross-correlation matrix analyses between the residues were carried out using the method described in our previous studies [25,33]. The positive areas (red) represent the correlation of residue motions, whereas the negative areas (blue) imply strong anticorrelated movement of residues [39]. The color of the diagonal portions in the matrixes exhibit the motion extent of the residues [32,39]. As shown in Figure 7, similar behaviors could be observed in the residues of diagonals, particularly residues 132–143 (corresponding to residues 132–143 of _A_PD-L1 of C-terminal, showing stronger movements) which is consistent with the previous RMSF results. However, differences in the movement of residues occurred between the four systems during the timescale of the simulations [39]. As expected, anticorrelated motions of the residues were observed in the matrix of the dimer system, while both EGCG and FRA weakened the anticorrelated movements in some regions. Notably, the anticorrelated motions in the FRA_EGCG system are heavily reduced compared to the EGCG and FRA systems. In general, these phenomena characterize the tight binding of the PD-L1 dimer to the small molecules, especially for FRA_EGCG. Along with the binding free energy calculations, the ability of EGCG to stabilize the PD-L1 dimer was markedly improved when FRA was introduced.

### 2.8. Free Energy Landscape

To further explore the conformational space of the PD-L1 dimer stabilized by these small molecules, their free energy landscapes, which were constructed on the basis of the projection of the first two principal components (dihedral PC1 and PC2), were comparatively analyzed (Figure 8) [40]. As shown in Figure 8a,b, four major zones are recognized for the EGCG system, while more major zones of conformations are discriminated for the PD-L1 dimer in the absence of small molecules, manifesting that EGCG can attenuate the conformational change in the PD-L1 dimer. By comparison, the PD-L1 dimer in the presence of FRA has three main zones. This demonstrates that FRA can reduce one more conformational zone than EGCG, suggesting FRA may have a stronger ability to stabilize the PD-L1 dimer than EGCG. However, FRA_EGCG can reduce two more conformational zones than EGCG and one more conformational zone than FRA, showing that FRA_EGCG has the strongest ability to stabilize the PD-L1 dimer. Hence, we can infer that the introduction of FRA significantly strengthens the capability of EGCG to inhibit the PD-1/PD-L1 pathway, which matches well with the calculated binding free energies. 

### 2.9. Secondary Structure

Final analysis of the secondary structure was conducted by using the dictionary of secondary structure for proteins (DSSP) program implemented in GROMACS and the results are depicted in Figure 9. As presented in Figure 9a, the secondary structures of the dimer system predominantly consisted of β-sheet conformations, such as the residues 37–42, 93–100, and 104–114 (corresponding to the residues 54–59, 110–117, and 121–123 of _A_PD-L1). The residues 32–35 (corresponding to the residues 49–52 of _A_PD-L1) possess the conformation of 3-helix throughout the simulation period but also slight deviations of turn conformation, while the residues 158–161 (corresponding to the residues 49–52 of _B_PD-L1) showed opposite behaviors. The residues 72–77 and 198–204 (corresponding to the residues 89–94 of _A_PD-L1 and 89–95 of _B_PD-L1, respectively) obtained conformational transitions of 3-helix, α-helix and turned throughout the entire simulation. The residues 116–126 (corresponding to the residues 133–143 of _A_PD-L1) exhibited the conformation of α-helix until 84 ns but was later transformed into turn, 3-helix, and coil conformations. The residues 242–249 (corresponding to the residues 133–140 of _B_PD-L1) also possess these four conformations throughout the simulation, suggesting that the C-terminal is not stable.

Similar phenomena were observed in the secondary structure compositions of the complex systems (Figure 9b–d), where the β-sheet conformation was well preserved in the 150 ns simulations, although it also shifted somewhat as the simulation time was prolonged. In particular, the C, F, and G sheet regions are very much essential for drug discovery, since they are critical areas that bind to PD-1. Thus, these compounds could stably interact with the PD-L1 dimer, thereby interrupting the PD-1/PD-L1 pathway. These findings could provide guidance for discovering two or more natural inhibitors with an additive effect on this pathway.

## 3. Materials and Methods

### 3.1. Preparation of the Initial Structures and Molecular Docking

The initial three-dimensional (3D) structures of FRA (PDB ID: 6NM7), EGCG (PDB ID: 3NG5), and the PD-L1 dimer in complex with BMS-202 (PDB ID: 5N2F) were downloaded from the Protein Data Bank (PDB). The missing structure of the PD-L1 dimer was completed by the WHAT IF online server and was loaded into AutoDock Vina 1.5.6 software for subsequent docking. The optimized geometries of the small molecules were obtained from Chem 3D 19.1.1.21 software in the MM2 forcefield. Then, they were docked into the PD-L1 dimer and a 20 × 20 × 20 Å^3^ box was chosen as the docking region, in which the center was set as the center of the binding site and all other parameters were maintained as the default [6,35,36]. Be similar to the reported literatures, the complex conformations with the lowest binding energy, named EGCG and FRA systems, were selected as the starting structures for MD simulations. The FRA_EGCG system was formed by placing FRA into the EGCG system directly and the results were visualized via PyMOL 2.5. The same procedure described in our previous studies was used to validate the docking parameters [25].

### 3.2. Molecular Dynamics Simulation

MD simulations were carried out with the binding conformations obtained from molecular docking utilizing the GROMACS 2016.4 software [41]. Amber ff99SB force field [42] and generalized AMBER force fields (GAFF) [43] were applied to parameterize the PD-L1 dimer and small molecules, respectively. All complex systems were solvated by TIP3P water models in cubic boxes with periodic boundary conditions, and the distance between the complexes and the box wall was set as 10 Å. An appropriate number of counterions were then added to ensure the entire system at pH 7.0. Subsequently, system energy minimizations were performed with both the steepest descent and conjugate gradient methods. After minimization, the systems were gently heated from 0 to 300 K over 1 ns using a Berendsen thermostat. The NPT ensemble was conducted using a Parrinello–Rahman barostat at 1 atmospheric pressure and 300 K. Finally, 150 ns MD simulations were performed forty times per the system for the production phase, during which the bonds involving hydrogen atoms were constrained using the LINCS algorithm. For calculation of the long-range electrostatic interactions, the Particle Mesh Ewald (PME) approach was used. The atomic coordinates were saved every picosecond for subsequent analysis.

### 3.3. Binding Free Energy Calculation

The binding free energies (Δ*G*) approximated by the difference between averaged free energies of the protein–ligand complex, the unbound protein, and the free ligand were calculated using the MM-PBSA approach (Equation (1)) [44]. Conformational sampling was performed every 100 ps from the last 30 ns trajectories of each simulation. Thus, 300 snapshots of the simulated structure of the stable MD trajectories were used for the calculation. Moreover, according to the definition of binding free energy, *G* was divided into three components, the gas-phase binding free energy (*E*_MM_), solvation free energy (*G*_sol_), and entropic contribution (–*T*Δ*S*) (Equation (2)). The *E*_MM_ is made up of the electrostatic energies (*E*_ele_) and van der Waals interaction energies (*E*_vdw_) (Equation (3)). The *G*sol consists of the electrostatic contribution (*E*_PB_) and nonpolar contribution to the solvation free energy (*E*_SA_) (Equation (4)). *E*_PB_ is calculated by the Poisson–Boltzmann (PB) model and *E*_SA_ is determined by the solvent accessible surface area with γ set as default (Equation (5)). To save the computational cost, the entropic contribution upon ligand binding was not considered [45,46]. Therefore, the relative binding free energies were evaluated in the present work to determine the binding affinities between the PD-L1 dimer and small molecules (Equation (6)). All data are expressed as means ± standard deviation of different simulations.
(1)$${\Delta G=G}_{\mathrm{complex}\;}{-\;(G}_{\mathrm{protein}\;}{\;+\;G}_{\mathrm{ligand}})$$
(2)$${G=E}_{\mathrm{MM}}{\;+\;G}_{\mathrm{sol}}\;-\;T\Delta S$$
(3)$$E_{\mathrm{MM}}{=E}_{\mathrm{vdw}}{\;+\;E}_{\mathrm{ele}}$$
(4)$$G_{\mathrm{sol}}{=E}_{\mathrm{PB}}{\;+\;E}_{\mathrm{SA}}$$
(5)$$E_{\mathrm{SA}}=\gamma \cdot \mathrm{SASA}$$
(6)$${\Delta G=\Delta E}_{\mathrm{vdw}}{\;+\;\Delta E}_{\mathrm{ele}}{\;+\;\Delta E}_{\mathrm{PB}}{\;+\;\Delta E}_{\mathrm{SA}}$$

To unveil the residues that are important for binding to small molecules, calculation of the per-residue free energy contribution was also performed. The residues with a binding free energy contribution of more than −1 kcal/mol were considered as key contributors for binding. Details about the calculation are in accordance with the former reports by our team [25,33].

### 3.4. Trajectory Analysis

The utility tools implemented in GROMACS 2016.4 were applied for the trajectory analysis of each simulated system. The DSSP program was employed to identify the secondary structure of the PD-L1 dimer [41]. The contact numbers and non-bonded interactions were computed using gmx mindist program and Protein–Ligand Interaction Profiler (PLIP), respectively [25,33]. The occupancies of intermolecular hydrogen bonds (H bonds) were analyzed using visual molecular dynamics (VMD) 1.9.3 software with a common standard [47]. In order to observe the internal dynamic properties of the PD-L1 dimer, an effective technique principal component analysis (PCA) was applied [48,49,50]. In the present study, the eigenvectors (i.e., the principal components) and eigenvalues were obtained by the diagonalization of covariance matrix of Cα atomic fluctuations. Then, the first two principal components (dihedral PC1 and PC2) were served as reaction coordinates to build the free energy landscapes [51,52,53,54].

## 4. Conclusions

In this work, MD simulations were applied to explore the effect of FRA on the binding ability of EGCG to the PD-L1 dimer. The results of conformational dynamics indicated that the FRA_EGCG system possessed more stable behavior during the entire simulation. Analysis of the binding free energy and per-residue energy decomposition revealed that the FRA_EGCG system displayed remarkable binding affinity compared with the EGCG system (−37.51 ± 0.43 kcal/mol vs. −20.31 ± 0.35 kcal/mol). This may be mainly manifested in two aspects, emerging key residues _A_Tyr56, _A_Met115, _B_Tyr123, _A_Ile54, and the increased energy contributions of existing key residues _A_Tyr123 and _B_Val68. The binding mode and non-bonded interaction results implied that FRA_EGCG mainly binds to C-, F- and G-sheet regions of the PD-L1 dimer. Notably, introduction of FRA predominantly enhanced the nonpolar interactions of the FRA_EGCG system. The free energy landscape and secondary structure analyses further suggested that FRA_EGCG can stably interact with the PD-L1 dimer throughout the simulation. In general, such structural features will help to identify two or more promising natural products for inhibiting the PD-1/PD-L1 axis collectively.

## Figures and Tables

**Figure 1 molecules-28-07881-f001:**
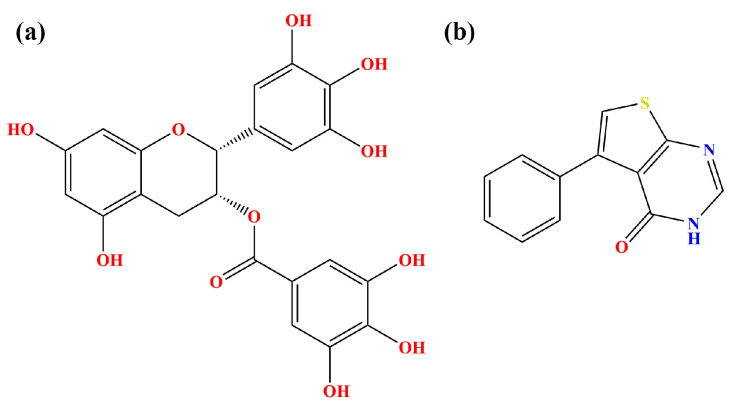
Structural formulas of (**a**) EGCG and (**b**) FRA.

**Figure 2 molecules-28-07881-f002:**
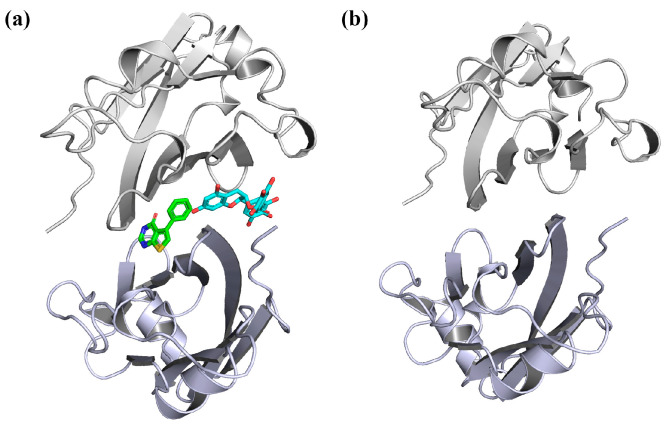
Initial structures of the systems used in MD simulations. (**a**) EGCG, FRA and FRA_EGCG systems. (**b**) Dimer system. The compounds EGCG and FRA are displayed in cyan and green, respectively. The atoms O, N and S are colored in red, blue and yellow, respectively.

**Figure 3 molecules-28-07881-f003:**
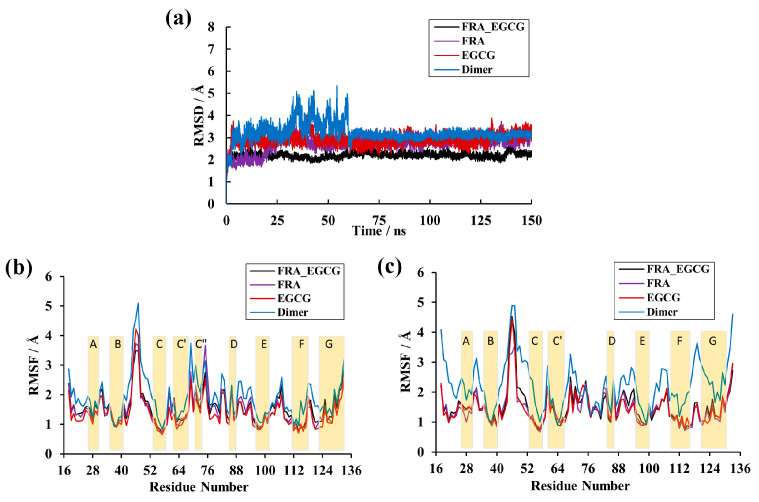
RMSD and RMSF results of the MD simulations. (**a**) RMSDs of the EGCG, FRA, FRA_EGCG, and dimer systems. (**b**) RMSF fluctuations of residues on _A_PD-L1. (**c**) RMSF fluctuations of residues on _B_PD-L1. The letters A–G represent the β-sheet domain.

**Figure 4 molecules-28-07881-f004:**
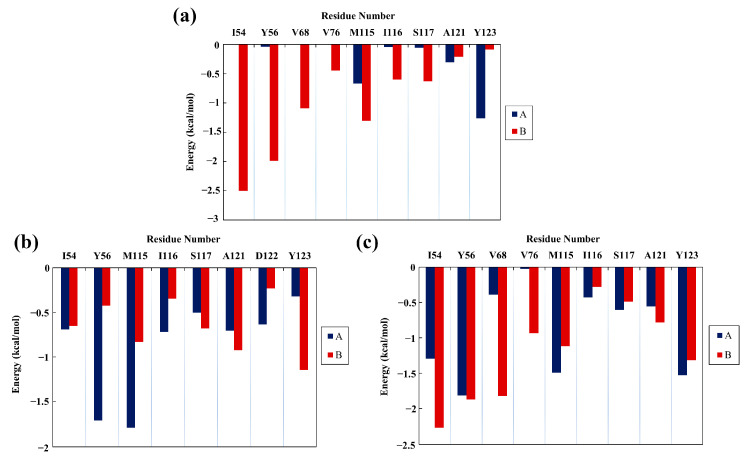
Energy contributions of key residues in the (**a**) EGCG, (**b**) FRA, and (**c**) FRA_EGCG systems (kcal/mol).

**Figure 5 molecules-28-07881-f005:**
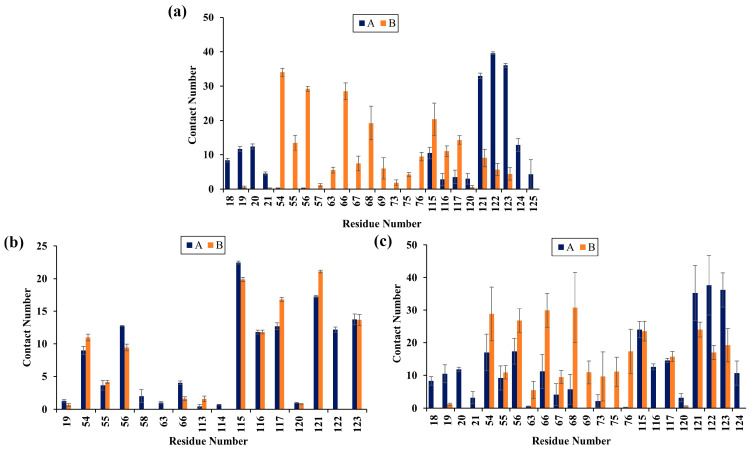
Contact numbers of (**a**) EGCG, (**b**) FRA, and (**c**) FRA_EGCG with the PD-L1 dimer. Error bars represent the standard deviations of triplicate calculations.

**Figure 6 molecules-28-07881-f006:**
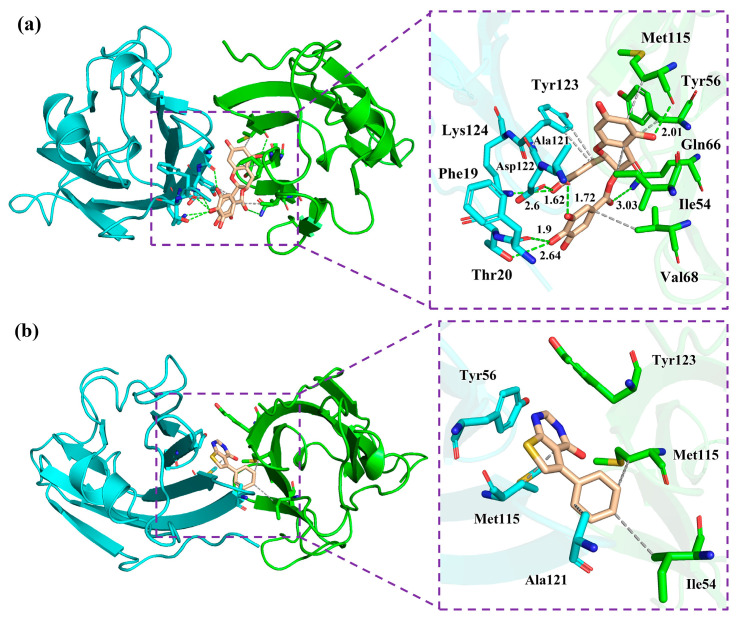
Detail of binding modes of the (**a**) EGCG, (**b**) FRA, and (**c**) FRA_EGCG systems. The residues on _A_PD-L1 and _B_PD-L1 at the binding pocket are shown as cyan and green sticks, respectively, while the ligands are shown as beige sticks. Hydrophobic interactions and H bonds are displayed as grey and green dashes, respectively. The atoms O, N and S are colored in red, blue and yellow, respectively.

**Figure 7 molecules-28-07881-f007:**
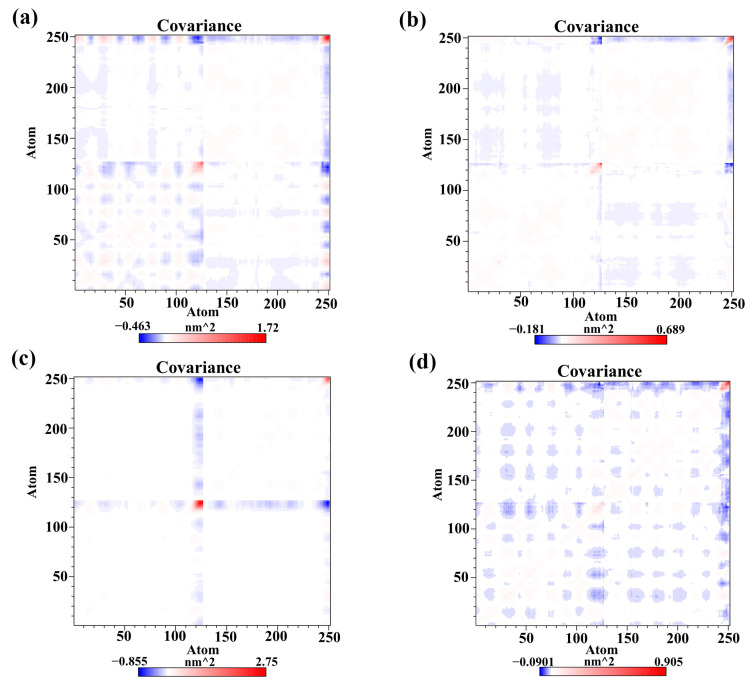
Cross-correlation matrixes of fluctuations for Cα atoms belonging to the PD-L1 dimer in the (**a**) EGCG, (**b**) FRA, (**c**) FRA_EGCG, and (**d**) dimer systems.

**Figure 8 molecules-28-07881-f008:**
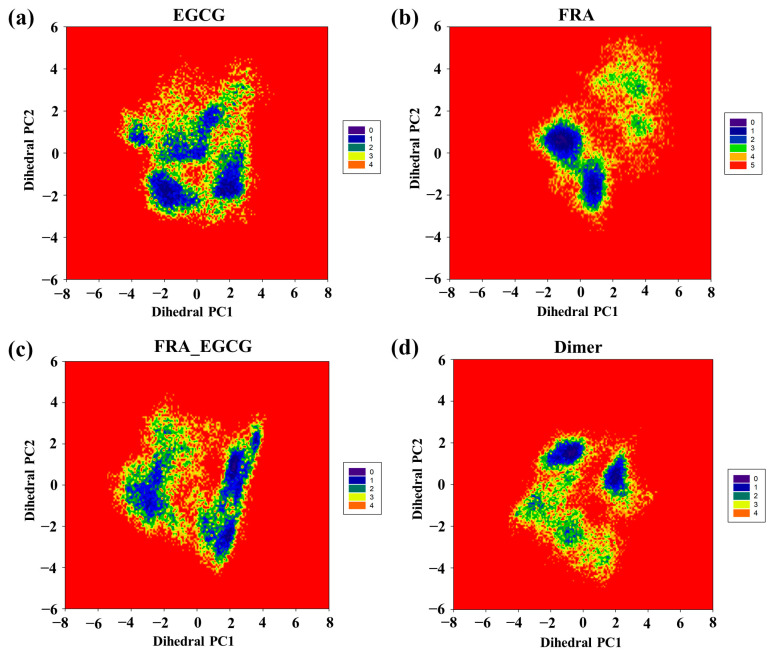
Free energy landscapes (KT) of the (**a**) EGCG, (**b**) FRA, (**c**) FRA_EGCG, and (**d**) dimer systems, reprinted from ref. [25]. PC1 and PC2 represent the largest two principal components.

**Figure 9 molecules-28-07881-f009:**
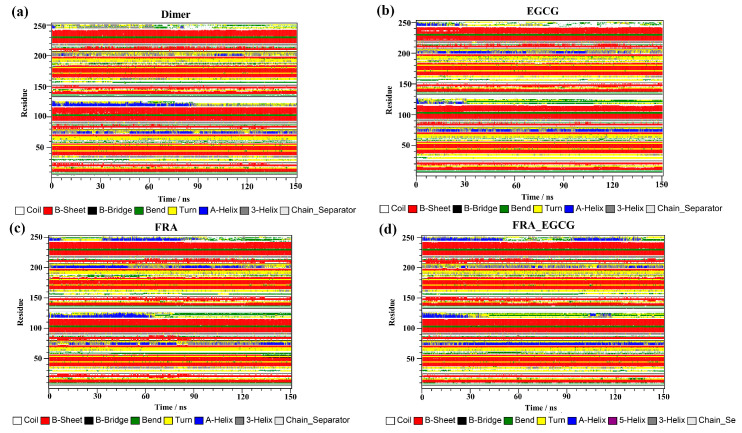
Evolution of secondary structures for the PD-L1 dimer over time in the (**a**) dimer, (**b**) EGCG, (**c**) FRA, and (**d**) FRA_EGCG systems.

**Table 1 molecules-28-07881-t001:** Binding free energies in the EGCG, FRA, FRA_EGCG and dimer systems (kcal/mol).

Contribution	EGCG	FRA	FRA_EGCG	Dimer
Δ*E*_vdw_ ^a^	−42.88 ± 1.70	−37.14 ± 0.51	−74.75 ± 2.19	−44.59 ± 9.85
Δ*E*_ele_ ^b^	−23.14 ± 3.14	−3.43 ± 0.13	−32.17 ± 2.29	−124.35 ± 23.36
Δ*E*_PB_ ^c^	50.43 ± 3.76	19.43 ± 0.83	76.89 ± 2.94	211.28 ± 17.07
Δ*E*_SA_ ^d^	−4.73 ± 0.16	−3.14 ± 0.02	−7.48 ± 0.28	−6.23 ± 0.37
Δ*E*_polar,total_ ^e^	27.29 ± 1.42	16.00 ± 0.97	44.72 ± 2.31	86.94 ± 10.00
Δ*E*_nonpolar,total_ ^f^	−47.61 ± 1.75	−40.28 ± 0.53	−82.23 ± 2.35	−50.82 ± 10.06
Δ*G* ^g^	−20.31 ± 0.35	−24.27 ± 0.27	−37.51 ± 0.43	36.11 ± 0.89

^a^ Van der Waals interaction energy. ^b^ Electrostatic energy. ^c^ Polar solvent effect energy. ^d^ Nonpolar solvent effect energy. ^e^ Polar binding free energy. ^f^ Nonpolar binding free energy. ^g^ Binding free energy. The energies are the average values of the 300 conformations extracted from 120 ns to 150 ns.

**Table 2 molecules-28-07881-t002:** H bond occupancies of the EGCG system.

Donor	Donor H	Acceptor	Occupancy (%)
EGCG@O50	H50	_A_Ala121@O	80.40
EGCG@O10	H10	_A_Asp122@N	59.80
EGCG@O47	H47	_A_Phe19@O	59.47
EGCG@O10	H10	_A_Asp122@OD1	39.53
EGCG@O03	H03	_B_Met115@O	43.52

**Table 3 molecules-28-07881-t003:** H bond occupancies of the FRA_EGCG system.

Donor	Donor H	Acceptor	Occupancy (%)
EGCG@O10	H10	_A_Asp122@OD2	100.00
EGCG@O50	H50	_A_Asp122@OD2	93.38
EGCG@O47	H47	_A_Thr20@OG1	23.84
_A_Lys124@NZ	HZ1	EGCG@O10	17.55
EGCG@O03	H03	_B_Gln66@OE1	80.64

## Data Availability

The data presented in this study are contained within the article.

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
