# Peer review of "Effect of Fragment 1 on the Binding of Epigallocatechin Gallate to the PD-L1 Dimer Explored by Molecular Dynamics"

_molecules, 2023, doi:10.3390/molecules28237881_

Round 1
Reviewer 1 Report
Comments and Suggestions for Authors
The paper by Guo and colleagues describes the computational study of the EGCG and 5-phenylthieno[2,3-d]pyrimidin-4(3H)-one (abbreviated as Fragment 1) as inhibitors of PD-L1/PD-1 interaction. The manuscript is well-organized and well-written. The methodology seems to be solid. However, there are two major issues, that should be addressed before the acceptance of the paper for publication. For instance:
1) It is hardly possible, that the administration of tested compounds would have an additive effect since Fragment 1 has an affinity in a mM range. The authors should merge Fragment 1 and EGCG into one molecule and compare the results with their work presented in the paper.
2) The results should be discussed in terms of the hot spots on the surface of the PD-L1 (see Kitel et al., ACS Chem. Biol. 2022, 17, 2655). The location of each fragment of the merged molecule should be indicated.
Reviewer 2 Report
Comments and Suggestions for Authors
The study titled "Effect of Fragment 1 on the Inhibitory Capacity of Epigallocate-chin Gallate to PD-L1 Dimerization: A Molecular Dynamics Simulation Study" provides a comprehensive overview in the field of cancer immunotherapy. The biomarker selection is of quite importance, as blockade of PD-1/PD-L1 interaction is a promising approach to restoring T cells for removing cancer cells. Overall, the theme was significant; however, a few suggestions need to be addressed, such as:
Title: very long and traditional. Authors should make it more impressive by enhancing it according to the weight of their research work. An impressive research topic should serve as the foundation for your work; guiding your research, focusing on specific questions, narrowing the literature review, and organizing your study should be considered when devising a topic.
Abstract: Unfortunately, the present research's objectives were not clear. Please remove unnecessary details from this section and make it more prominent. Revise lines 19–25 to make it more attractive for global readers. The conclusion present in the last few lines was very weak. Please improve. Please use a standardized way of mentioning amino acids throughout the manuscript. Lines 27–29 need revision.
Introduction: This section has many details, which, in my opinion, need a thorough revision. Kindly use updated literature to support your introduction. The mentioned details should be aligned with the study's rationale and scope of work. What is the justification for mentioning lines 46–50? Lines 76–94: The authors should justify why they were emphasizing such details. In my opinion, these lines will be more suitable for the discussion section. The introduction has been written without thorough literature. With careful attention and a comprehensive literature review, kindly improve this section and make it simpler for readers. At the end of this section, please revise your objective according to the study's scope.
Results and Discussion: The impression of starting results and discussion simply from 2.1. Root-mean-squared The deviation did not look well. It will be better to write the discussion separately. Please justify why RMSD was computed at 150 ns. Line 108–112, please re-check these details. The authors should elaborate in their discussion about why the variations happened between 40 and 52 residues, specifically by the DIMER (Figure 3 b and c). The non-significant findings of Table 1 should be discussed accordingly. What is the purpose of mentioning lines 234–236? Please justify. What is the significance of mentioning Table 2 details? Please discuss. Discuss in detail the inference from figure 6. Please describe the impact of tables 3 and 4 on the study’s rationale. What is the prominent significance of Figure 7? Please discuss it in detail. Can you please reduce the number of figures? Thus, you will need to make effective use of the supplementary information and include only essential or relevant tables and figures in your paper proper.
Materials and methods: Please add a paragraph in a more comprehensive way for the readers to track the protein details, so it will ensure its future reproducibility. Line 373, please justify why you mentioned “data not shown”. Please complete and improve heading 3.2. Please re-check heading 3.4.
Conclusion: Please revise and reduce the unnecessary details in this section and connect them with future prospects as well.
References: Few are old; if possible, kindly mention references from 2016–2023 onwards.
Good luck!
Comments on the Quality of English Language
Moderate editing of English language required
Round 2
Reviewer 1 Report
Comments and Suggestions for Authors
The revised manuscript can be accepted for publication.
Author Response
We accept your comments, and thank you very much for your kind cooperation!
Thank you very much again for your kind comments and suggestions!
Reviewer 2 Report
Comments and Suggestions for Authors
All the suggested changes are now satisfactorily addressed in the revised manuscript.
Author Response

(The authors gave the same response as above.)
